# Real-World Efficacy of Glucagon-like Peptide-1 (GLP-1) Receptor Agonist, Dulaglutide, on Metabolic Parameters in Japanese Patients with Type 2 Diabetes: A Retrospective Longitudinal Study

**DOI:** 10.3390/biomedicines11030869

**Published:** 2023-03-13

**Authors:** Hisayuki Katsuyama, Mariko Hakoshima, Shohei Umeyama, Sakura Iida, Hiroki Adachi, Hidekatsu Yanai

**Affiliations:** Department of Diabetes, Endocrinology and Metabolism, National Center for Global Health and Medicine Kohnodai Hospital, Chiba 272-8516, Japan

**Keywords:** GLP-1 receptor agonist, dulaglutide, type 2 diabetes

## Abstract

The glucagon-like peptide-1 receptor agonist (GLP-1RA) dulaglutide has been shown to improve body weight and glycemic control and reduce major cardiovascular (CV) events. In Japan, dulaglutide is used at a fixed dose of 0.75 mg, which is lower than that in Europe and North America. However, the reports of real-world efficacy on metabolic parameters in Japanese patients with type 2 diabetes (T2DM) are limited. This study aimed to examine the real-world efficacy of GLP-1RA dulaglutide on metabolic parameters in Japanese patients with T2DM. We retrospectively selected patients with T2DM who had been prescribed dulaglutide continuously for 12 months or longer between September 2015 and December 2020 and compared metabolic parameters at baseline with the data at 12 months after the start of dulaglutide. One hundred twenty-one patients were enrolled in this study. The 12-month dulaglutide treatment reduced body weight by 1.7 kg and hemoglobin A1c by 1.1%. Significant improvements were also observed in serum high-density lipoprotein cholesterol (HDL-C), triglyceride (TG) and non-HDL-C. The change in HbA1c during dulaglutide treatment was significantly correlated with the changes in HDL-C (R = −0.236, *p* = 0.013), LDL-C (R = 0.377, *p* = 0.005) and non-HDL-C (R = 0.415, *p* < 0.001). The improvements in HbA1c, HDL-C, TG and non-HDL-C were greater in patients concurrently treated with SGLT2 inhibitors (SGLT2is) at baseline. In conclusion, the treatment with dulaglutide has beneficial effects on multiple CV risk factors in Japanese patients with T2DM.

## 1. Introduction

The prevalence of diabetes is increasing worldwide, and the situation is critical in Asia. It was estimated that 90 million adults in Southeast Asia suffered from diabetes in 2010, and the number was expected to reach 113 million by 2030 [1]. Despite recent developments in pharmacotherapy, the life expectancy of diabetes was shorter than the general population [2].

One of the leading causes of mortality in patients with diabetes is cardiovascular (CV) disease. CV diseases accounted for 14.9% of deaths in Japanese patients with diabetes [2]. Large-scale trials suggested the importance of interventions against CV risk factors such as hyperglycemia, dyslipidemia and obesity to prevent CV diseases in patients with diabetes [3].

Glucagon-like peptide-1 (GLP-1) is a peptide hormone that is secreted by enteroendocrine cells and promotes insulin secretion from pancreatic beta cells. Dulaglutide is one of the human GLP-1 receptor agonists (GLP-1RAs), administered as a once-weekly subcutaneous injection, and is used worldwide for the treatment of type 2 diabetes (T2DM) [4]. Previous reports revealed that dulaglutide improved glycemic control with a lower risk of hypoglycemia and was associated with weight loss [5]. Moreover, in a large-scale randomized placebo-controlled study (REWIMD trial), dulaglutide reduced three-point major adverse CV events in patients with T2DM with a high risk of CV diseases [6].

Although dulaglutide is used at a fixed dose of 0.75 mg in Japan, which is lower than the doses that range from 0.75 mg to 4.5 mg in Western countries, improvements in glycemic control and obesity by dulaglutide were observed in Japanese patients with T2DM in several clinical trials [7,8,9]. However, the number of reports on real-world efficacy in Japanese patients with T2DM is limited. The aim of this study is to examine the effects of dulaglutide on various metabolic conditions such as hyperglycemia, obesity, dyslipidemia, diabetic nephropathy and metabolic-associated fatty liver disease.

## 2. Materials and Methods

### 2.1. Study Population

We selected patients with T2DM who had been prescribed dulaglutide continuously at a fixed dose of 0.75 mg for 12 months or longer between 1 September 2015 and 31 December 2020 by a chart-based analysis. We compared retrospectively the data at baseline with the data at 12 months after the dulaglutide treatment started.

### 2.2. Data Collection

The data on anthropometric measurements, blood tests, urine tests and medications were obtained from medical charts. The anthropometric measurements, such as body weight, height, waist circumference and blood pressure, were conducted according to the clinical standards. Body mass index (BMI) was calculated by body weight in kilograms divided by the square of the height in meters. The results of blood tests included both fasting and postprandial samplings. Plasma glucose was obtained using the hexokinase method. Serum hemoglobin A1c (HbA1c), serum creatinine, serum total cholesterol (TC) and serum triglyceride (TG) were measured by enzymatic assays. Serum low-density lipoprotein cholesterol (LDL-C) and serum HDL-C were determined by a direct method. Serum aspartate aminotransferase (AST), serum alanine aminotransferase (ALT) and serum γ-glutamyl transferase (γGTP) were measured by the Japan Society of Clinical Chemistry transferable method. Urinary albumin was measured by turbidimetric immunoassay, and the albumin-to-creatinine ratio (ACR) was calculated. The estimated glomerular filtration rate (eGFR) was calculated using the following formula: eGFR = 194 × (serum creatinine − 1.094) × (age − 0.287) × (0.739, if female) [10]. Non-HDL-C was calculated as the difference between TC and HDL-C. Hepatic steatosis index (HSI) was calculated as 8 × (ALT/AST) + BMI + (2, if diabetes mellitus) + (2, if female) [11]. NAFLD fibrosis score (NFS) was calculated as −1.675  +  0.037 × age (years) + 0.094 × body mass index (BMI) (kg/m^2^) + 1.13 × DM (yes = 1, no = 0) + 0.99 × AST/ALT ratio − 0.013 × platelet count (×10^9^/L) − 0.66 × albumin (g/dL) [12]. FIB-4 index was calculated as a marker of hepatic fibrosis, using the following formula: (age × AST)/(platelet counts (× 10^9^/L) × (ALT)^1/2^ [13,14].

### 2.3. Statistical Analysis

Comparison of the variables before and after the dulaglutide treatment was analyzed by the Wilcoxon signed-rank test. Pearson’s simple correlation coefficients were performed to determine the correlations between the parameters. All data are expressed as mean ± SD, and *p* < 0.05 was considered to be statistically significant. Statistical analysis was conducted using SPSS version 23 (IBM, US).

## 3. Results

### 3.1. Baseline Characteristics of Patients Studied

We found 197 patients who had been first prescribed dulaglutide between 1 September 2015 and 31 December 2020. Among them, 25 patients did not visit our hospital regularly. Five patients died during the observational period due to cancer (n = 3), cerebral infarction (n = 1) and pneumonia (n = 1). In 22 patients, dulaglutide prescription was suspended and switched to other antihyperglycemic agents in 18 patients (insulin 1, other GLP-1RAs 2, dipeptidyl peptidase-4 (DPP-4) inhibitors 13, metformin 1 and SGLT2 inhibitor 1). We also excluded 24 patients due to lack of sufficient data. Thus, we enrolled 121 patients in this study.

The baseline characteristics of the patients are presented in Table 1. Dulaglutide was administered at a dose of 0.75 mg in all patients. The mean age of the patients was 64.7 ± 15.6 years, and the mean BMI was 26.8 ± 5.7 kg/m^2^. Among the hypoglycemic agents, metformin was mostly used (47.9%), followed by SGLT2 inhibitors (43.0%) and Thiazolidinedione (30.0%). Insulin was used in 28 patients (23.1%). The most prescribed drug for hypertension was calcium channel blockers (47.9%) followed by angiotensin II receptor blockers (41.3%). Of all patients, 54.5% received statins. Antiplatelet drugs were prescribed in 18 patients (14.9%).

### 3.2. Changes in Metabolic Parameters during 12-Month Dulaglutide Parameters

Changes in metabolic parameters at 12 months after the start of dulaglutide are shown in Table 2. HbA1c decreased by 1.1%, and the body weight decreased by 1.7 kg during 12-month dulaglutide therapy. Significant improvements were also observed in systolic blood pressure, plasma glucose, serum albumin, HDL-C, TG and non-HDL-C. Serum γGTP and NFS were also improved, whereas HSI and FIB-4 index were not changed during the dulaglutide treatment. There were also no significant changes in eGFR, uric acid and ACR.

Table 3 shows the gender differences in the changes in metabolic parameters during the 12-month dulaglutide treatment. Significant improvements in plasma glucose, HbA1c, γGTP and NFS were observed in both male and female patients. Body weight and BMI were improved significantly in males, and there were the same tendencies in females. Serum TG and non-HDL-C levels decreased only in males, whereas HDL-C levels increased only in females.

### 3.3. Correlations between the Baseline and the Changes in Metabolic Parameters

The significant correlations between the baseline values and the changes during 12 months of dulaglutide treatment were observed in NFS (R = −0.265, *p* = 0.007), the FIB-4 index (R = −0.325, *p* < 0.001) and ACR (R = −0.473, *p* < 0.001) (Figure 1). The same tendency was also observed in HSI (R = −0.224, *p* = 0.064).

### 3.4. Correlations among the Changes in Metabolic Parameters

The correlations between the changes in metabolic parameters during the 12-month dulaglutide treatment are provided in Table 4. The changes in HbA1c were significantly correlated with the changes in HDL-C (R = −0.236, *p* = 0.013), LDL-C (R = 0.377, *p* = 0.005) and non-HDL-C (R = 0.415, *p* < 0.001). Nevertheless, the change in BMI was not correlated with the changes in HbA1c, TC, LDL-C and non-HDL-C. The change in HSI was correlated with the change in BMI, but not with the change in NFS or the FIB-4 index.

### 3.5. Subgroup Analysis in Patients with or without Insulin Treatment

We analyzed the changes in metabolic parameters in the subgroups with or without insulin treatments (Table 5), since insulin has anabolic effects and is associated with weight gain [15]. In patients with insulin treatment, significant decreases were observed only in HbA1c, TC, non-HDL-C and NFS, whereas there were significant increases in diastolic blood pressure and uric acid levels. In patients without insulin treatment, there were significant improvements in body weight, BMI, systolic blood pressure, plasma glucose, HbA1c, γGTP, HDL-C, uric acid and NFS.

### 3.6. Subgroup Analysis in Patients with or without SGLT2i Treatment

Since SGLT2 inhibitors were previously reported to reduce body weight and improve glycemia, dyslipidemia and liver function [16,17,18], we divided the patients into two subgroups under the treatments with or without SGLT2 inhibitors and analyzed the changes in metabolic parameters during the 12-month dulaglutide treatment (Table 6). Significant improvements in body weight, plasma glucose, HbA1c and NFS were observed in both groups. TC, HDL-C, TG and non-HDL-C were improved only in the patients with the treatment with SGLT2i. There were no significant changes in HSI, the FIB-4 index and ACR in both groups.

## 4. Discussion

In this study, we examined the real-world efficacy of once-weekly subcutaneous dulaglutide on metabolic parameters in Japanese patients with T2DM. The dulaglutide treatment was associated with improvement in obesity, glycemia and atherogenic lipid profile.

Reductions from baseline in HbA1c and body weight in dulaglutide-treated patients were consistent with previous clinical trials. It was reported that the 52-month dulaglutide treatment with a single oral hypoglycemic agent in Japanese patients with T2DM decreased HbA1c by 1.57% to 1.69% [7]. The other study also showed a reduction in HbA1c by 1.39% after the 52-month dulaglutide monotherapy [8]. In our study, the change in HbA1c during dulaglutide treatment was 1.1%, which was slightly smaller compared to such previous clinical trials. The higher age of the patients and multiple concomitant hypoglycemic agents in our study might influence the smaller degree of HbA1c changes.

Asian patients with T2DM were characterized by relatively lower BMI [19]. A previous study clarified that the degree of HbA1c change was greater during GLP-1RA treatment in normal BMI patients compared to obese patients [20]. It was also reported that GLP-1RAs were more effective in lowering HbA1c in Asian patients with T2DM compared to Caucasians [21]. Our results confirmed that dulaglutide effectively reduced HbA1c in Asian patients with T2DM.

GLP-1RAs have beneficial effects not only on glycemia and obesity but also on lipid profiles. Previous randomized controlled studies including REWIND or TODAY trials reported reductions in serum LDL-C and TG levels [6,22,23]. Moreover, a meta-analysis of four trials revealed that GLP-1RAs improved serum LDL-C, TG and HDL-C levels [24]. In our study, dulaglutide significantly increased serum HDL-C levels and decreased TG and non-HDL-C levels. A tendency for a reduction in LDL-C levels was also observed. These results coincided with previous clinical trials. In other several real-world studies, liraglutide or exenatide also showed improvements in carotid intima–media thickness, suggesting the preventive effects of GLP-1RAs against CV diseases [25,26].

Interestingly, the changes in LDL-C, HDL-C and non-HDL-C were significantly associated with the change in HbA1c, which suggests a close association between the improvements of glycemia and lipid profiles by dulaglutide. In an animal study using ApoE knockout mice, both glycemic control and lipid profile were improved but only observed in STZ-induced diabetic model mice and not in non-diabetic mice [27]. The presence of diabetes or insulin resistance elevates the expression and activity of hormone-sensitive lipase in adipose tissue, which catalyzes lipolysis and the release of free fatty acid (FFA). Increased entry of FFA to the liver inhibits the degradation of apoB-100 and elevated production of very-low-density lipoprotein (VLDL). Insulin resistance reduces the activity of lipoprotein lipase (LPL), which, in turn, increases TG-rich lipoproteins such as intermediate-density lipoprotein (IDL) and VLDL and reduces HDL. These are observed as elevated serum TG levels and decreased HDL-C levels in clinical laboratory tests [28]. Improved insulin action by GLP-1RA can play a crucial role in ameliorating both glycemia and dyslipidemia. GLP-1RAs can promote postprandial insulin secretion and inhibit insulin-mediated lipolysis in adipose tissue, resulting in reductions in FFA release in the bloodstream and entry in the liver [29,30]. GLP-1RAs were also reported to improve insulin resistance [31], which elevates the activity of LPL and ameliorates the lipid profile. Indeed, reductions in VLDL-C levels by GLP-1 RA treatment were already reported [32,33]. Reductions in apoB-100 comprising lipoproteins including IDL, VLDL and LDL can be observed as the improvement in non-HDL-C levels [34], which was observed in our study.

Previous studies reported the beneficial effects of GLP-1RAs in the progression of non-alcoholic fatty liver disease (NAFLD) [35,36]. In our study, dulaglutide treatments improved only NFS, but there were no changes in HSI and the FIB-4 index. Nevertheless, the associations between baseline values and changes in HSI, NFS and the FIB-4 index suggested that dulaglutide could ameliorate steatosis and fibrosis in the liver in patients with advanced steatosis and fibrosis at baseline. The lower dose of dulaglutide used in Japan or the limited number of subjects might influence our results.

The use of GLP-1RAs was also associated with reduced ACR and the risk of worsening kidney function [37,38]. In our study, there were no significant changes in ACR in the whole group or even in the group with SGLT2i treatment, despite the renoprotective effects of SGLT2is [39]. Our results might be affected by various factors, such as the lack of a control group, the limited number of available data on ACR, the short period of observation and the large standard deviation of ACR. Further studies with a control group and longer observation will be necessary to examine the effect of dulaglutide on the progression of NAFLD or diabetic kidney diseases.

Previous reports suggested that the addition of dulaglutide to insulin contributed to better glycemia, smaller doses of insulin and smaller weight gain compared to insulin monotherapy [40]. The combination of insulin and GLP-1RAs was also associated with better lipid profiles and lower blood pressure [41]. These reports suggested that GLP-1RAs might offset the adverse effects of insulin treatments. In our study, dulaglutide improved HbA1c and non-HDL-C in patients treated with insulin, but there were no significant changes in body weight and BMI, which might be influenced by the small number of patients with insulin treatments. Direct comparisons with insulin monotherapy will be needed to determine the effects of the combination therapy of insulin and dulaglutide.

SGLT2is have also been reported to improve obesity, glycemia and lipid profiles and were associated with reduced cardiorenal events [16,17,42]. The mechanisms of cardiorenal effects of SGLT2is include attenuation of CV risk factors, amelioration of substrate utilization in myocardial and renal cells, reduction in ventricular and vascular loading through diuresis and natriuresis, and mitigation of atherosclerosis through anti-inflammatory and anti-oxidative effects [42,43]. GLP-1RAs have also been reported to preserve ventricular function and reduce atherosclerosis [44]. These effects of SGLT2is and GLP-1RAs can be complementary in the prevention of CV diseases. Indeed, a meta-analysis revealed that the combination therapy of SGLT2i and GLP-1RA resulted in greater reductions in body weight, HbA1c and LDL-C [45]. Our real-world data confirmed the greater benefits of the combination of SGLT2i and GLP-1RA on glycemia and lipid profiles, suggesting a promising strategy to prevent the progression of CV and renal diseases in high-risk patients [46,47].

Our study has several limitations. First, it was a retrospective study with a limited number of patients in a single center. Second, we cannot exclude possible influences by modifications of the pharmacotherapy. Third, our study lacked information on diabetic durations, co-existing diabetic complications and lifestyle modifications. Fourth, a lack of data might influence the results. Fifth, we could not evaluate metabolites or inflammation markers which might be related to the effects of dulaglutide, since our study was based on real-world data.

## 5. Conclusions

Our findings based on real-world data showed that dulaglutide improved body weight, glycemic control and lipid profiles in Japanese patients with T2DM, suggesting dulaglutide as a promising therapy to control CV risk factors. Moreover, the close associations between the changes in lipid profiles and the change in HbA1c during dulaglutide treatments is intriguing in speculating the presence of common mechanisms of improving glycemia and dyslipidemia by dulaglutide.

## Figures and Tables

**Figure 1 biomedicines-11-00869-f001:**
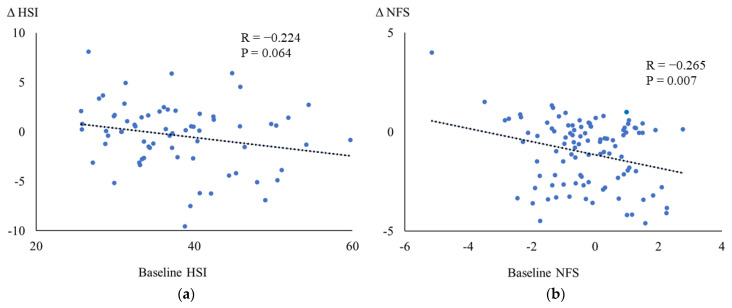
The results of Pearson’s simple correlations among the parameters during 12 months after the start of dulaglutide. (**a**) The correlation between the changes in HSI and the baseline HSI. (**b**) The correlation between the changes in NFS and baseline NFS. (**c**) The correlation between the changes in FIB-4 index and the baseline FIB-4 index. (**d**) The correlation between the changes in ACR and the baseline ACR. ACR, albumin-to-creatinine ratio; HSI, hepatic steatosis index; NFS, NAFLD fibrosis score.

**Table 1 biomedicines-11-00869-t001:** Baseline characteristics of the patients (n = 121).

Age	64.7 ± 15.6
Gender (M/F)	69/52
Body height (cm)	162 ± 10
Body weight (kg)	68.6 ± 19.2
Body mass index (BMI) (kg/m^2^)	26.8 ± 5.7
Medications at baseline	
Insulin	28 (23.1%)
Metformin	58 (47.9%)
Sulfonylurea	21 (17.3%)
Glinides	10 (8.3%)
Thiazolidinedione	36 (30.0%)
Alpha-glucosidase inhibitors	14 (11.6%)
SGLT2 inhibitors	51 (42.1%)
ACE Inhibitors	1 (0.8%)
Angiotensin II receptor blockers	50 (41.3%)
Calcium channel blockers	58 (47.9%)
Diuretics	14 (11.6%)
Alpha blockers	1 (0.8%)
Beta Blockers	10 (8.3%)
Statins	66 (54.5%)
Ezetimibe	15 (12.4)
Fibrates	8 (6.6%)
Antiplatelet drugs	18 (14.9%)

**Table 2 biomedicines-11-00869-t002:** Change in the clinical parameters 12 months after the start of dulaglutide (n = 121).

	n	Baseline	12 Months	*p*
Body weight (kg)	74	69.6 ± 21.0	67.9 ± 20.6	0.001
BMI (kg/m^2^)	74	26.9 ± 6.4	26.2 ± 6.5	0.001
Systolic blood pressure (mmHg)	67	129 ± 22	117 ± 30	0.027
Diastolic blood pressure (mmHg)	50	74 ± 12	75 ± 12	0.641
Plasma glucose (mg/dL)	119	201 ± 88	174 ± 78	<0.001
HbA1c (%)	116	8.8 ± 1.6	7.7 ± 1.4	<0.001
Albumin (g/dL)	108	4.04 ± 0.59	4.14 ± 0.50	0.017
AST (IU/L)	117	29 ± 19	28 ± 19	0.359
ALT (IU/L)	118	35 ± 31	32 ± 27	0.301
γGTP (IU/L)	105	67 ± 127	53 ± 74	0.001
Total bilirubin	53	0.66 ± 0.33	0.64 ± 0.34	0.628
TC (mg/dL)	104	183 ± 47	176 ± 41	0.075
HDL-C (mg/dL)	114	49 ± 13	51 ± 15	0.011
LDL-C (mg/dL)	58	101 ± 32	94 ± 30	0.086
TG (mg/dL)	115	199 ± 226	168 ± 155	0.026
Non-HDL-C (mg/dL)	103	135 ± 48	126 ± 39	0.017
Creatinine (mg/dL)	121	0.85 ± 0.43	0.88 ± 0.43	0.373
eGFR (mL/min/1.73 m^2^)	121	73 ± 28	70 ± 25	0.170
Uric acid (mg/dL)	102	5.3 ± 1.8	5.1 ± 1.4	0.148
Hemoglobin (g/dL)	119	13.4 ± 2.0	13.5 ± 1.8	0.279
Platelet (×10^4^/μL)	119	22.9 ± 7.2	22.7 ± 7.2	0.489
HSI	69	37.8 ± 7.9	37.4 ± 7.8	0.630
NFS	101	−0.28 ± 1.35	−1.34 ± 1.87	<0.001
FIB-4 index	112	1.72 ± 1.17	1.69 ± 1.11	0.591
ACR (mg/g Cre)	56	202 ± 569	183 ± 502	0.408

Values show mean ± SD. ACR, albumin-to-creatinine ratio; ALT, alanine aminotransferase; AST, aspartate aminotransferase; BMI, body mass index; eGFR, estimated glomerular filtration rate; γGTP, γ-glutamyl transferase; HbA1c, hemoglobin A1c; HDL-C, high-density lipoprotein cholesterol; HSI, hepatic steatosis index; LDL-C, low-density lipoprotein cholesterol; NFS, NAFLD fibrosis score; TC, total cholesterol; TG, triglyceride.

**Table 3 biomedicines-11-00869-t003:** Change in the clinical parameters 12 months after the start of dulaglutide in male and female patients.

	Male (n = 69)				Female (n = 52)			
	n	Baseline	12 Months	*p*	n	Baseline	12 Months	*p*
Body weight (kg)	42	74.2 ± 23.5	72.3 ± 23.2	0.005	32	63.6 ± 15.5	62.1 ± 14.7	0.055
BMI (kg/m^2^)	42	26.6 ± 6.8	25.9 ± 6.9	0.006	32	27.2 ± 5.9	26.7 ± 6.1	0.059
Systolic blood pressure (mmHg)	39	130 ± 21	122 ± 31	0.302	28	128 ± 23	109 ± 28	0.033
Diastolic blood pressure (mmHg)	31	74 ± 13	76 ± 12	0.428	19	73 ± 11	73 ± 10	0.879
Plasma glucose (mg/dL)	69	201 ± 81	184 ± 85	0.012	50	200 ± 97	162 ± 64	0.004
HbA1c (%)	68	8.8 ± 1.5	7.8 ± 1.5	<0.001	48	8.8 ± 1.7	7.6 ± 1.3	<0.001
Alb (g/dL)	61	4.10 ± 0.58	4.17 ± 0.54	0.135	47	3.95 ± 0.58	4.09 ± 0.44	0.057
AST (IU/L)	66	30 ± 20	27 ± 18	0.355	51	29 ± 17	29 ± 20	0.718
ALT (IU/L)	67	37 ± 31	31 ± 24	0.082	51	32 ± 30	33 ± 31	0.645
γGTP (IU/L)	61	80 ± 92	64 ± 92	0.044	44	47 ± 47	39 ± 33	0.007
Total bilirubin (mg/dL)	34	0.74 ± 0.36	0.71 ± 0.37	0.504	52	0.72 ± 0.37	0.79 ± 0.39	0.968
TC (mg/dL)	61	184 ± 50	169 ± 43	0.003	43	182 ± 41	185 ± 36	0.579
HDL-C (mg/dL)	66	47 ± 13	48 ± 12	0.397	48	52 ± 12	57 ± 17	0.007
LDL-C (mg/dL)	34	101 ± 31	95 ± 27	0.452	24	102 ± 33	93 ± 34	0.113
TG (mg/dL)	67	213 ± 271	170 ± 182	0.019	48	180 ± 139	166 ± 108	0.427
Non-HDL-C (mg/dL)	60	139 ± 52	123 ± 42	0.004	43	130 ± 41	129 ± 34	0.772
Creatinine (mg/dL)	69	0.95 ± 0.41	0.95 ± 0.44	0.504	52	0.72 ± 0.37	0.79 ± 0.39	0.026
eGFR (mL/min/1.73 m^2^)	69	73 ± 28	72 ± 24	0.875	52	73 ± 29	68 ± 27	0.052
Uric acid (mg/dL)	56	5.6 ± 1.7	5.4 ± 1.4	0.352	46	5.0 ± 1.8	4.8 ± 1.3	0.260
Hemoglobin (g/dL)	68	13.8 ± 2.2	13.9 ± 2.2	0.505	51	12.9 ± 1.7	13.0 ± 1.5	0.435
Platelet (×10^4^/μL)	68	22.1 ± 7.4	22.3 ± 7.7	0.673	52	23.9 ± 6.6	23.2 ± 6.5	0.110
HSI	39	37.2 ± 8.1	36.3 ± 8.2	0.063	30	38.6 ± 7.5	38.9 ± 7.0	0.131
NFS	56	−0.34 ± 1.42	−1.39 ± 1.87	<0.001	45	−0.20 ± 1.26	−1.08 ± 1.66	0.001
FIB-4 index	63	1.78 ± 1.35	1.74 ± 1.25	0.859	49	1.66 ± 0.88	1.64 ± 0.91	0.547
ACR (mg/g Cre)	31	172 ± 352	145 ± 390	0.468	25	238 ± 755	229 ± 610	0.716

Values show mean ± SD. ACR, albumin-to-creatinine ratio; ALT, alanine aminotransferase; AST, aspartate aminotransferase; BMI, body mass index; eGFR, estimated glomerular filtration rate; γGTP, γ-glutamyl transferase; HbA1c, hemoglobin A1c; HDL-C, high-density lipoprotein cholesterol; HSI, hepatic steatosis index; LDL-C, low-density lipoprotein cholesterol; NFS, NAFLD fibrosis score; TC, total cholesterol; TG, triglyceride.

**Table 4 biomedicines-11-00869-t004:** The correlations between the changes in metabolic parameters during 12-month dulaglutide treatment.

	ΔBMI	ΔHbA1c	ΔTG	ΔHDL-C	ΔLDL-C	ΔNon-HDL-C	ΔHSI	ΔNFS
ΔBMI	1							
ΔHbA1c	0.112	1						
ΔTG	−0.021	0.165	1					
ΔHDL-C	0.245 *	−0.236 *	−0.121	1				
ΔLDL-C	0.097	0.377 **	−0.233	0.215	1			
ΔNon-HDL-C	0.046	0.398 **	0.415 **	−0.040	0.518 **	1		
ΔHSI	0.586 **	0.037	0.064	0.141	0.208	−0.026	1	
ΔNFS	0.201	0.067	0.268 **	0.013	−0.083	0.149	−0.212	1
ΔFIB-4 index	0.059	−0.008	−0.005	0.060	−0.094	0.076	−0.197	−0.474 **

* *p* < 0.05; ** *p* < 0.01; BMI, body mass index; eGFR, estimated glomerular filtration rate; HbA1c, hemoglobin A1c; HDL-C, high-density lipoprotein cholesterol; HSI, hepatic steatosis index; LDL-C, low-density lipoprotein cholesterol; NFS, NAFLD fibrosis score; TG, triglyceride.

**Table 5 biomedicines-11-00869-t005:** Change in the clinical parameters 12 months after the start of dulaglutide in the subgroups with or without insulin treatment at baseline.

	With Insulin at Baseline (n = 28)				Without Insulin at Baseline (n = 93)			
	n	Baseline	12 Months	*p*	n	Baseline	12 Months	*p*
Body weight (kg)	15	71.5 ± 21.3	71.0 ± 20.6	0.572	59	69.1 ± 21.0	67.1 ± 20.5	0.001
BMI (kg/m^2^)	15	26.9 ± 6.4	26.8 ± 6.5	0.638	59	26.9 ± 6.4	26.1 ± 6.5	0.001
Systolic blood pressure (mmHg)	15	128 ± 23	134 ± 30	0.346	52	130 ± 22	112 ± 28	0.003
Diastolic blood pressure (mmHg)	13	74 ± 10	82 ± 11	0.023	37	74 ± 13	72 ± 11	0.362
Plasma glucose (mg/dL)	28	188 ± 75	172 ± 83	0.183	91	205 ± 92	175 ± 76	<0.001
HbA1c (%)	27	9.0 ± 1.7	8.1 ± 1.3	0.008	89	8.8 ± 1.6	7.6 ± 1.4	<0.001
Alb (g/dL)	25	4.0 ± 0.7	4.0 ± 0.7	0.695	83	4.0 ± 0.5	4.2 ± 0.4	0.010
AST (IU/L)	26	24 ± 10	23 ± 9	0.833	91	31 ± 20	29 ± 21	0.358
ALT (IU/L)	27	30 ± 20	29 ± 18	0.964	91	36 ± 33	33 ± 29	0.274
γGTP (IU/L)	23	77 ± 158	63 ± 86	0.135	82	63 ± 117	51 ± 71	0.005
Total bilirubin (mg/dL)	15	0.68 ± 0.42	0.66 ± 0.35	0.893	38	0.66 ± 0.29	0.63 ± 0.33	0.593
TC (mg/dL)	23	183 ± 30	167 ± 24	0.032	81	183 ± 51	178 ± 44	0.440
HDL-C (mg/dL)	23	50 ± 13	51 ± 22	0.696	91	49 ± 12	52 ± 13	0.002
LDL-C(mg/dL)	10	103 ± 33	97 ± 381	0.508	48	101 ± 32	93 ± 30	0.104
TG (mg/dL)	24	212 ± 219	147 ± 65	0.241	91	196 ± 228	174 ± 171	0.069
Non-HDL-C (mg/dL)	22	135 ± 36	119 ± 25	0.025	81	135 ± 51	127 ± 42	0.113
Creatinine (mg/dL)	28	0.87 ± 0.24	0.92 ± 0.27	0.330	93	0.85 ± 0.45	0.87 ± 0.47	0.697
eGFR (mL/min/1.73 m^2^)	28	70 ± 26	63 ± 18	0.130	93	74 ± 29	73 ± 27	0.525
Uric acid (mg/dL)	22	4.8 ± 1.8	5.4 ± 1.4	0.032	80	5.4 ± 1.8	5.0 ± 1.3	0.006
Hemoglobin (g/dL)	28	13.4 ± 2.1	13.3 ± 1.9	0.202	91	13.4 ± 2.0	13.6 ± 1.8	0.056
Platelet (×10^4^/μL)	28	23.6 ± 7.0	24.9 ± 9.9	0.973	91	22.7 ± 7.2	22.0 ± 6.0	0.429
HSI	13	38.0 ± 7.3	38.5 ± 6.2	0.600	56	37.7 ± 8.0	37.2 ± 8.1	0.443
NFS	22	−0.34 ± 1.29	−1.70 ± 1.80	0.009	79	−0.26 ± 1.37	−1.24 ± 1.88	<0.001
FIB-4 index	25	1.51 ± 1.05	1.50 ± 1.00	0.968	87	1.79 ± 1.19	1.75 ± 1.13	0.537
ACR (mg/g Cre)	9	74 ± 123	81 ± 103	0.953	47	226 ± 616	202 ± 544	0.302

Values show mean ± SD. ACR, albumin-to-creatinine ratio; ALT, alanine aminotransferase; AST, aspartate aminotransferase; BMI, body mass index; eGFR, estimated glomerular filtration rate; γGTP, γ-glutamyl transferase; HbA1c, hemoglobin A1c; HDL-C, high-density lipoprotein cholesterol; HSI, hepatic steatosis index; LDL-C, low-density lipoprotein cholesterol; NFS, NAFLD fibrosis score; TC, total cholesterol; TG, triglyceride.

**Table 6 biomedicines-11-00869-t006:** Change in the clinical parameters 12 months after the start of dulaglutide in the subgroups with or without SGLT2i treatment at baseline.

	With SGLT2i at Baseline (n = 52)				Without SGLT2i at Baseline (n = 69)			
	n	Baseline	12 Months	*p*	n	Baseline	12 Months	*p*
Body weight (kg)	26	70.6 ± 17.7	69.7 ± 17.8	<0.001	48	69.0 ± 22.6	66.9 ± 21.9	0.004
BMI (kg/m^2^)	26	27.4 ± 5.3	27.1 ± 5.6	0.059	48	26.6 ± 6.9	25.8 ± 7.0	0.006
Systolic blood pressure (mmHg)	24	124 ± 19	118 ± 28	0.797	48	132 ± 23	116 ± 31	0.018
Diastolic blood pressure (mmHg)	19	73 ± 15	74 ± 12	0.777	31	74 ± 11	75 ± 12	0.600
Plasma glucose (mg/dL)	51	200 ± 84	171 ± 66	0.009	68	202 ± 92	177 ± 85	0.007
HbA1c (%)	50	9.2 ± 1.6	7.9 ± 1.2	<0.001	66	8.6 ± 1.6	7.6 ± 1.5	<0.001
Alb (g/dL)	48	4.1 ± 0.5	4.3 ± 0.4	0.003	60	4.0 ± 0.6	4.0 ± 0.5	0.458
AST (IU/L)	51	30 ± 22	29 ± 21	0.490	66	29 ± 17	27 ± 17	0.484
ALT (IU/L)	52	36 ± 34	35 ± 33	0.813	66	33 ± 28	29 ± 21	0.250
γGTP (IU/L)	43	89 ± 187	59 ± 95	0.007	62	51 ± 51	49 ± 56	0.060
Total bilirubin (mg/dL)	19	0.72 ± 0.35	0.68 ± 0.24	0.634	34	0.63 ± 0.32	0.62 ± 0.38	0.805
TC (mg/dL)	45	190 ± 56	180 ± 50	0.033	59	178 ± 37	173 ± 31	0.616
HDL-C (mg/dL)	49	47 ± 11	52 ± 12	<0.001	65	50 ± 13	51 ± 17	0.744
LDL-C (mg/dL)	28	98 ± 26	93 ± 28	0.269	30	104 ± 37	94 ± 32	0.136
TG (mg/dL)	50	250 ± 316	188 ± 214	0.002	65	160 ± 100	153 ± 82	0.638
Non-HDL-C (mg/dL)	44	143 ± 59	130 ± 49	0.006	59	129 ± 37	123 ± 29	0.483
Creatinine (mg/dL)	52	0.88 ± 0.47	0.86 ± 0.40	0.657	69	0.84 ± 0.35	0.89 ± 0.45	0.150
eGFR (mL/min/1.73 m^2^)	52	73 ± 26	72 ± 24	0.414	69	73 ± 30	69 ± 26	0.277
Uric acid (mg/dL)	48	5.4 ± 1.8	5.1 ± 1.4	0.022	54	5.2 ± 1.7	5.1 ± 1.3	0.866
Hemoglobin (g/dL)	50	14.0 ± 1.9	14.3 ± 1.7	0.048	69	12.9 ± 2.0	13.0 ± 1.7	0.980
Platelet (×10^4^/μL)	50	21.9 ± 6.6	22.1 ± 6.1	0.435	69	23.6 ± 7.5	23.1 ± 7.9	0.173
HSI	25	39.7 ± 7.5	39.6 ± 7.5	0.143	44	36.7 ± 7.9	39.2 ± 7.9	0.283
NFS	46	−0.21 ± 1.21	−1.63 ± 1.54	<0.001	55	−0.33 ± 1.46	−1.10 ± 2.08	0.035
FIB-4 index	49	1.74 ± 1.33	1.62 ± 1.10	0.088	63	1.71 ± 1.02	1.75 ± 1.12	0.547
ACR (mg/g Cre)	29	124 ± 323	93 ± 163	0.983	27	285 ± 740	279 ± 690	0.313

Values show mean ± SD. ACR, albumin-to-creatinine ratio; ALT, alanine aminotransferase; AST, aspartate aminotransferase; BMI, body mass index; eGFR, estimated glomerular filtration rate; γGTP, γ-glutamyl transferase; HbA1c, hemoglobin A1c; HDL-C, high-density lipoprotein cholesterol; HSI, hepatic steatosis index; LDL-C, low-density lipoprotein cholesterol; NFS, NAFLD fibrosis score; TC, total cholesterol; TG, triglyceride.

## Data Availability

The data supporting the findings of this study are available from the corresponding author upon reasonable request.

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
