# Peer review of "Real-World Efficacy of Glucagon-like Peptide-1 (GLP-1) Receptor Agonist, Dulaglutide, on Metabolic Parameters in Japanese Patients with Type 2 Diabetes: A Retrospective Longitudinal Study"

_biomedicines, 2023, doi:10.3390/biomedicines11030869_

Round 1

Reviewer 1 Report

Current manuscript submitted by Hisayuki K et al and titled as “Real-World Efficacy of GLP-1 receptor agonist, Dulaglutide, on Metabolic parameters in Japanese patients with T2D: A Retrospective Longitudinal Study”. The longitudinal study performed retrospectively in patients with T2D underwent treatment with GLP-1 receptor agonist. Authors determined the effects of GLP-1 receptor agonist Dulaglutide on metabolic parameters (high glucose, lipid profile and liver injury markers and injury score and urinary albuminuria etc) in Japanese patients with Type-2 Diabetes. The current results suggesting that treatment with Dulaglutide improved several metabolic profiles including body weight, hyperglycemia, lipid profile which associated with cardiovascular risk factors. The current study design is appropriate and written well. However, several concerns need to address.

Comments:

1) Authors could have also measured several metabolites related to uric acid cycle, nucleic acid metabolism Xanthine oxidoreductase (XOR), glycolysis related metabolites, amino acids, ketone bodies and inflammation marker. Some of them are predicted markers known to be observed in T2D patients and it is interesting that Dulaglutide had any effects?

2) I wonder authors did not comment about gender differences in metabolic profile before and after treatment with Dulaglutide?

3) Based on the results, Dulaglutide has beneficial effects by modulating the serum metabolic profile in patients with T2D including HbA1C is decreased. However, In Table 2, uACR (Albuminuria) did changed significantly after 12 months of Dulaglutide treatment? Explain?

4) In Table4, also uACR did not changed significantly even after treatment with SGLT2i? explain why? because there are many evidences suggesting that SGLT2i improved the uACR together with kidney histological changes.  

5) In Results, there is only one Figure. However, authors indicated as Figure 2. Correct it.  

Author Response

We are extremely grateful to you for your very careful review of our manuscript and for the suggestions that were instrumental in improving the quality of our manuscript.

1) This study was based on real-world data in clinical practice. Thus, it was difficult to evaluate metabolites or markers related to the effects of dulaglutide. We added this point to the limitation in the “Discussion”.

2) We added Table 3, which shows the gender differences in changes in metabolic parameters during the dulaglutide treatment. Overall, the effects of dulaglutide were similar in male and female patients.

3) We considered that various factors such as the lack of a control group, the limited number of available data on ACR, the short period of observation and the large standard deviation of ACR influenced our negative results. Further study with a control group and a longer observation period will be needed to examine the effects of dulaglutide in the progression of diabetic kidney diseases. We added this point to the “Discussion”.

4) As the reviewer pointed out, various studies confirmed the renoprotective effects of SGLT2is. SGLT2is had been already administered at the start of the dulaglutide treatment, and thus, it was difficult to observe changes in uACR even in the patient group.

5) We apologize for the mistake. We corrected it.

Reviewer 2 Report

Title: Real-world Efficacy of Glucagon-like Peptide-1 (GLP-1) Receptor Agonist, Dulaglutide, on Metabolic Parameters in Japanese Patients with Type 2 Diabetes: A Retrospective Longitudinal Study

Authors: Hisayuki Katsuyama, Mariko Hakoshima , Shohei Umeyama , Sakura Iida , Hiroki Adachi and Hidekatsu Yanai

General comment:

We have observed a true revolution in diabetes treatment in the recent years that was associated with the introduction of novel drugs such as GLP-1 receptor agonists and SGLT2 inhibitors. In several RCTs, these compounds proved efficacy not only in glycemic control but also in the prevention of macro and microvascular complications of diabetes. These findings constituted a basis for new guidelines for diabetes treatment. However, populations recruited to RCTs are highly selected and therefore may not correspond to the general population. Therefore real-world studies are of significant value as they help to verify drug efficacy in real clinical settings. In their work, Hisayuki Katsuyama evaluated the efficacy of a 1-year treatment with dulaglutide in real-world settings by a retrospective analysis of 121 patients with T2D. The study constitutes an important contribution to the research field; however, there are some issues the Authors should consider before the manuscript is accepted for publication.

Major revisions:

Section: Abstract

1)  Please add information on the dulaglutide dose.

2) "The improvements in HbA1c and lipid profiles were greater in patients concurrently treated with SGLT2 inhibitors (SGLT2is) at baseline." – please consider that SGLT2 inhibitors tend to increase LDL cholesterol levels. Importantly, in this study, the SGLT2i treated group did not notice an improvement in LDL cholesterol level, too. Therefore, I would suggest reorganizing this sentence in the abstract to make it consistent with the study findings.

Section: Study population

1)  Please add information on the dulaglutide dose and if the patients included in the study were encouraged to introduce any lifestyle modifications.

Section: Baseline characteristics of patients studied 

1)      Please add information on diabetes duration and co-existing complications

Section: Discussion

1)     If data on diabetes duration, co-existing diabetes complications, and lifestyle modifications is not available, please add this information to the study limitations.

2)     Please consider that lack of improvements in the context of liver steatosis may result from a relatively low dose of dulaglutide used in the study.

Minor revisions:

Section: Introduction

1) Page 1, line 30: “It was estimated that 90 million adults in South-East Asia suffered from diabetes…”

– please provide the year

2) Page 2, lines 4-5: “Despite a lower maximum dose of 0.75 mg in Japan compared to the higher dose of  1.5 mg in Western countries, improvements in glycemic control and obesity by dulaglutide were observed in Japanese patients with T2DM in several clinical trials [7-9].” – please take into consideration that in Europe and North America available doses of dulaglutide range from 0.75 to 4.5 mg. A dose of 0.75 mg is treated as an initial one and an increase of the dose to 1.5 mg is recommended in most patients. Therefore it is important to add information on the dulaglutide dose in the Abstract and Material and Methods.

Author Response

We are extremely grateful to you for your very careful review of our manuscript and for the suggestions that were instrumental in improving the quality of our manuscript.

Major revisions:

Section: Abstract

1) In Japan, dulaglutide is used at a fixed dose of 0.75 mg. We added this information to the “Abstract”, Introduction”, “Material and Methods” and “Results”.

2) As the reviewer pointed out, several studies reported that SGLT2i treatment was associated with a slight increase in LDL-C levels. In our study, there were no significant changes in LDL-C levels in patients with SGLT2i treatment. We corrected the text as follows, “The improvements in HbA1c, HDL-C, TG and non-HDL-C were greater in patients concurrently treated with SGLT2 inhibitors (SGLT2is) at baseline”.

Section: Study population

1) We added the information on the dulaglutide dose. It was a retrospective study, and there was no information on lifestyles. We added the possible influences of lifestyle modifications to one of the limitations of our study in the “Discussion”.

Section: Baseline characteristics of patients studied

1) The diabetic duration and co-existing complications could influence our results, but we did not have sufficient data. We added this point to the limitations in the “Discussion”

Section: Discussion

1) We added these points to the study limitations in the “Discussion”.

2) It has not been reported whether the effects of dulaglutide on the progression is dose-dependent. Nevertheless, as the reviewer pointed out, the smaller dose of dulaglutide might lead to lack of significant changes in the indexes of NAFLD, We added this point to the “Discussion".

Minor revisions:

Section: Introduction

1) It was the data in 2010. We added the year in the text.

2) In Japan, dulaglutide is used at a fixed dose of 0.75 mg. We added this information to the “Abstract”, Introduction”, “Material and Methods” and “Results”.

Reviewer 3 Report

I read the paper “Real-world Efficacy of Glucagon-like Peptide-1 (GLP-1) Receptor Agonist, Dulaglutide, on Metabolic Parameters in Japanese Patients with Type 2 Diabetes: A Retrospective Longitudinal Studyby Katsuyama et al.

The article is well written. Statistical analysis Is well performed.

Comment:

1.     This study has more limitations than the ones reported. It is a single centre experience, thus limiting the generalization of the results. In addition, the limited sample size is further limited by a huge amount of missing data for some of the parameters explored. Moreover, the concomitant use of other antihyperglycemic treatment is an important bias. The author reported the role of SGLT2i dividing the population in table 4. But also insulin therapy should be evaluated (present in 28 patients). In fact, it is well known that insulin acts as an anabolise, thus increasing weight gain.

2.     The combination therapy between SGLT2i and GLP1 RA is not completely discussed in the appropriate section. In fact, it is well known the role of SGLT2i in diabetic treatment, from the correction of cardiorenal risk factors, metabolic adjustments ameliorating myocardial substrate utilization, optimization of ventricular loading conditions through effects on diuresis, natriuresis, and vascular function and endothelial dysfunction (doi: 10.3390/ijms23073651) The potential of the combination therapy may enhance the value of the article. Please improve it.

3.     It is not clear to me weather all the patients enrolled had their diabetes therapy modified or not during the follow-up period. Moreover, please report, if possible, any change in the other drug therapy which could affect metabolism. For instance, 66 patients were on statin therapy at the beginning of the study and had a certain cholesterol, if we dismiss/modify this therapy or if we add other drugs to other patients or to these patients, I expect a change in cholesterol panel.  This could represent another bias.

Author Response

We are extremely grateful to you for your very careful review of our manuscript and for the suggestions that were instrumental in improving the quality of our manuscript.

  1. We added these points (single center, small sample size, missing data) to the limitations of our study in the “Discussion”. We also added Table 5, which shows the changes in metabolic parameters in patients with and without insulin treatments. In our study, only HbA1c and non-HDL-C levels were improved in the group with insulin treatment. Comparisons with insulin monotherapy will be needed to determine the effects of the combination therapy of insulin and GLP-1RAs. We discussed these points in the “Discussion”.

  1. As the reviewer pointed out, SGLT2is prevent cardiorenal diseases through various mechanisms and these effects might be complementary with those of GLP-1RAs. We added these points in the “Discussion”.

  1. Due to the nature of retrospective studies based on real-world data, it was difficult to avoid the influences of the modifications, but the modifications in pharmacotherapy were limited. For example, at the end of the observation period, 67 patients took statins (66 patients at baseline), and 52 patients took SGLT2is (51 patients at baseline). Our results did not change significantly, even if we exclude the patients, who were newly given statins or SGLT2is. Nevertheless, the possible influences of the treatment modifications were one of the important limitations of our study. We added this point to the limitations in the “Discussion”.

Round 2

Reviewer 1 Report

Congratulations to authors. All the comments raised by reviewers were addressed properly and appropriate. The current version of manuscript can be accepted to publication.

Reviewer 2 Report

I want to express my gratitude for the opportunity to re-review the paper entitled: "Real-world Efficacy of Glucagon-like Peptide-1 (GLP-1) Receptor Agonist, Dulaglutide, on Metabolic Parameters in Japanese Patients with Type 2 Diabetes: A Retrospective Longitudinal Study” by Hisayuki Katsuyama et al. Since the authors addressed all my concerns regarding the methodology and the manuscript structure, I find it acceptable for publishing in the “Biomedicines."

Reviewer 3 Report

The paper is much improved and can be further processed for publication. There are only few English mistakes to be solved (Abstract: use America and not Amerika; page 2, line 9 efficacy and not ef-ficacy; page 2 line 11 obesity and not ob-esity). Please double check the manuscript.